# USB: A Unified Summarization Benchmark Across Tasks and Domains

**Kundan Krishna**♣    **Prakhar Gupta**♣    **Sanjana Ramprasad**◇
**Byron C. Wallace**◇    **Jeffrey P. Bigham**♣    **Zachary C. Lipton**♣

♣ Carnegie Mellon University
◇Northeastern University

{kundank,prakharg,jbigham,zlipton}@andrew.cmu.edu

{ramprasad.sa,b.wallace}@northeastern.edu

## Abstract

While the NLP community has produced numerous summarization benchmarks, none provide the rich annotations required to simultaneously address many important problems related to control and reliability. We introduce a Wikipedia-derived benchmark, complemented by a rich set of crowd-sourced annotations, that supports 8 interrelated tasks: (i) extractive summarization; (ii) abstractive summarization; (iii) topic-based summarization; (iv) compressing selected sentences into a one-line summary; (v) surfacing evidence for a summary sentence; (vi) predicting the factual accuracy of a summary sentence; (vii) identifying unsubstantiated spans in a summary sentence; (viii) correcting factual errors in summaries. We compare various methods on this benchmark and discover that on multiple tasks, moderately-sized fine-tuned models consistently outperform much larger few-shot prompted language models. For factuality-related tasks, we also evaluate existing heuristics to create training data and find that training on them results in worse performance than training on $20\times$ less human-labeled data. Our articles draw from 6 domains, facilitating cross-domain analysis. On some tasks, the amount of training data matters more than the domain where it comes from, while for other tasks training specifically on data from the target domain, even if limited, is more beneficial. [1]

## 1 Introduction

Automatic text summarization has been an important, active research sub-area in NLP for over two decades (Radev et al., 2002; Nenkova et al., 2011; El-Kassas et al., 2021). Numerous summarization benchmarks have been proposed to facilitate the development of summarization methods (Nallapati et al., 2016; Narayan et al., 2018; Wang and Ling, 2016; Gliwa et al., 2019). However, the majority of previous work has primarily focused on evaluating the models' ability to generate summaries similar to reference summaries, neglecting key auxiliary properties of text summarization systems.

Recent research has highlighted the importance of addressing additional aspects in text summarization. These aspects include the ability to steer summaries by controlling its focus on a topic or on specific parts of the source text (Gehrmann et al., 2019). Furthermore, there is an increasing emphasis on ensuring *factual correctness* and implementing mechanisms to eliminate factual errors from model outputs (Scialom et al., 2021; Balachandran et al., 2022). Similarly, to foster trust in the outputs, it is desirable for summarization systems to present evidence from sources that corroborate the generated summaries. As models have improved in generating coherent and readable summaries (Goyal et al., 2022), these auxiliary considerations have gained importance. Aligning summaries with user requirements and ensuring sufficient factual support are critical frontiers in summarization research. The current summarization benchmarks fail to provide a comprehensive evaluation of model capabilities across various summarization tasks, encompassing properties such as factuality and controllability.

In this work, we introduce USB, a comprehensive benchmark for text summarization that supports eight auxiliary tasks. The benchmark includes labeled datasets with high-quality human annotations collected from diverse documents across six domains. To create the benchmark, we sampled Wikipedia articles from various categories, such as people, organizations, and events. We utilized the introductory section of the articles as a reference summary and the remaining content as the source text, resulting in imperfect source-"summary" pairs. Human annotators then searched for evidence to support each summary sentence. If evidence was lacking, corresponding spans or entire sentences

---

[1]The dataset can be downloaded from https://github.com/kukrishna/usb

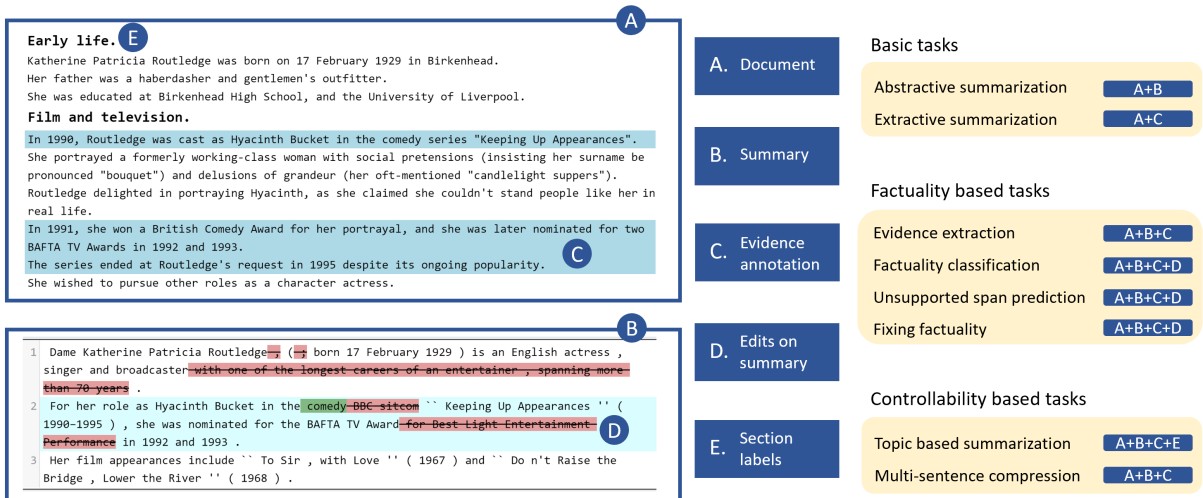

Figure 1: A schematic of our dataset, annotations, and the supported tasks. The example shown (abridged) displays the edits made by a human annotator on the initial candidate summary (deletions in red with strike-through; additions in green). Every summary sentence is supported by one or more evidence sentences highlighted in blue.

were removed. Whenever conflicting evidence was encountered, the summary was revised with minimal edits to align with the available evidence. The resulting annotations can be repurposed to create labeled datasets for 8 useful tasks (Figure 1).

We offer the first human-labeled training datasets for various summarization tasks, including evidence extraction and identifying spans in summaries without supporting evidence. These datasets enable the training and evaluation of models specifically for these crucial aspects. We benchmark the performance of several models such as instruction-tuned encoder-decoder models and LLMs on our tasks, including both fine-tuning and well as few-shot prompting based approaches. Notably, we found that fine-tuning even small models (fewer than a billion parameters) substantially outperforms few-shot prompting of much larger open-source and private large language models.

Prior efforts have relied on heuristics to generate synthetic training data for certain tasks included in our benchmark. For instance, a common heuristic employed is lexical overlap to identify and extract supporting evidence (Chen and Bansal, 2018). Similarly, artificial factual errors have been introduced into summaries to train models for factuality classification or correction (Kryściński et al., 2020; Balachandran et al., 2022). Although such automatic approaches enable easy creation of large training datasets, heuristically derived annotations are typically noisy compared to human annotations. Our findings demonstrate that models trained on minimal amount of human-labeled data outperform

those trained on heuristically generated labeled datasets, even when the latter are 20× larger.

A common challenge to real-world adoption of models is their use in resource-poor domains where one does not have access to abundant labeled training data. We compare how the size of available training data matters vis-a-vis its domain for different summarization tasks. We found that for tasks related to factual correctness of summaries, the amount of training data matters more than its domain; but for other tasks having domain-specific training data matters more. Our benchmark is explicitly segmented into 6 domains based on Wikipedia categories, and hence provides a natural test-bed for such domain transfer studies.

**Summary of contributions:**

- Multi-domain benchmark for training and evaluating models on 8 different tasks dealing with some critical but understudied aspects of text summarization.

- Comprehensive evaluation of models and training strategies, including fine-tuning, few-shot prompting, and multi-task training.

- Comparison of relative value of training data labels generated by humans versus heuristics, showing that for multiple tasks, human annotations yield better models even with 20× less training data.

- Practical insights about out-of-domain generalization for different tasks, identifying the

tasks for which the size of the training data matters more than it being from a specific target domain.

## 2  Dataset Curation

To create the USB benchmark, we first collected a set of manual annotations on Wikipedia articles. We then used the collected annotations to create labeled data for the benchmark tasks. In this section we describe the process of collecting these manual annotations. We consider the text in a Wikipedia article overview (leading) section as the target summary $S$, and the rest of the article as $D$. In well-written articles, the overview section ($S$) provides a broad summary of the article, and the rest of the article ($D$) provides specifics. Hence, the content in $S$ which highlights parts of $D$ can be effectively considered its summary. However, for $S$ to be a valid summary of $D$, we need to remove contents within it that mention *new* information that is not present in $D$ and cannot be inferred from it.

We recruited annotators and asked them to execute the following tasks: (1) Find and annotate evidence in $D$ for each summary sentence of $S$, and; (2) Delete parts of $S$ that are not supported by $D$. This yields a document-summary pair where the summary is fully supported by the document, and the supporting evidence is explicitly marked. We provide a detailed description of our data creation process below.

**Retrieval of Wikipedia articles**   We downloaded the Wikipedia English articles dump from 1 July 2022. We extracted the articles from this corpus using the Wikiextractor tool. [2] We dropped tables and lists during extraction, but retained section headers. We used a set of category filters to retrieve pages about specific types of entities which helps us in creating a dataset with diverse domains. We manually filtered domains to select those in which articles generally had a substantial part of $S$ supported by evidence present in $D$. We retrieved articles for the following 6 domains: biographies, companies, schools, newspapers, landmarks, and disasters.

**Selecting documents for annotation**   Our heuristic is to assume that the overview section of a Wikipedia article will feature a significant amount of overlap with the remaining part which would be retained after the annotators remove non-

overlapping parts. To derive a good document-summary pair from an article, there should ideally be a large amount of overlap between the overview part and remaining article. Otherwise, after human annotation (to remove parts of the summary unsupported by the corresponding document) one would be left with little text in the summary.

Given an article, with the overview section represented by $S$ and the remaining part represented by $D$, we broke the summary into sentences $s_1 s_2 s_3 ... s_n$ using Spacy[3]. We calculated how many of the summary sentences have at least one entity which is also present in $D$. For this step, we automatically marked entities in $S$ and $D$ by considering all the words with internal hyperlinks to other Wikipedia pages as entities. If two hyperlinked words pointed to the same page, they were considered the same entity. For annotation, we only retained articles that have more than $75\%$ of sentences in $S$ with at least one entity overlapping with $D$. We also controlled for summary length by discarding any article where $S$ has fewer than 4 or more than 12 sentences.

**Flagging entity overlaps to help annotators find evidence**   To help annotators find evidence supporting any given summary sentence, our interface highlights entities present in that sentence and also in the source document, with a different color for each entity. To maximize the number of entities detected, we took a union of entities extracted using Wikipedia's hyperlinks, Spacy and DBpedia. [4]

**Selection and monitoring of Mechanical Turk Workers**   We ran a qualification task on Mechanical Turk, tasking workers with annotating one document-summary pair according to the provided instructions. To take this qualifier, we required workers have a HIT approval rate $> 95\%$, and have more than 1000 approved HITS. Each worker was allowed to take the qualification task only once. All workers were given the same document-summary pair for annotation. A total of 174 workers took the qualification task. Out of these, 28 workers were approved by checking their annotation quality manually. The approved workers were then permitted to work on the *main* task where they were asked to annotate different document-summary pairs. Each pair was annotated by exactly one worker. After 300 annotations for the main task, we analyzed

---

[2] https://github.com/attardi/wikiextractor

[3] https://spacy.io
[4] https://www.dbpedia-spotlight.org

the annotation quality of the responses again. For many approved workers, the annotation quality on the main task was significantly worse than the qualification task, and hence we restricted the worker set to only 3 workers whose annotation quality was much better than the rest (hereafter referred to as *primary* workers). The remaining annotations were done by these workers, and a total of 1988 document-summary pairs were annotated.

**Verifying annotations**  Due to the complexity of the annotation task, evidence has not been annotated in some parts in the summaries after the first round. To address this, we trained a model to predict unsupported spans in summaries. Specifically, we trained models that accept an initial summary sentence $s$ and the evidence annotated by the workers as the input, and then predict which spans in $s$ were deleted by the annotator to in their submitted version $s'$. We applied this model to the summary sentences submitted by annotators to predict unsupported spans in them. We fine-tuned Flan-T5 XL (Chung et al., 2022) for this task. We divided the set of document-summary pairs annotated by our primary workers into two halves, trained a model on each half, and used it to predict the unsupported spans in the other half. We used one of these models for prediction on the remaining document-summary pairs submitted by other workers. Using these model predictions, we selected around 20% of the total summary sentences most likely to contain unsupported spans, and flagged them for verification. This included about 15% of the sentences annotated by primary workers and 45% of sentences annotated by other workers, which aligns with our manual inspection of quality of the workers' annotations. We then designed a slightly modified interface for the verification task, where summary sentences have highlights showing potentially unsupported content, and the workers can select additional evidence or edit the summary as before. After incorporating the changes made in this verification round, we arrived at the final version of the annotated corpus.

## 3   Task Definitions

We derived labeled datasets for tasks using the collected annotations. The resulting benchmark consists of the following 8 tasks:

**Extractive Summarization (EXT):** Given the full document as input, extract all important sentences that it contains. We define the ideal "reference" extractive summary as the set of all source sentences marked as evidence for the summary.

**Abstractive Summarization (ABS):** Generate a multi-sentence summary of the source document by not just simply selecting important content, but also rephrasing it for succinctness and coherence. The ground truth is the full-length summary created after the annotators' edits.

**Factuality Classification (FAC):** Predict if a summary sentence is factually correct and sufficiently supported by the information present in the source. We create labeled data by assigning *non-factual* and *factual* labels to the before and after versions of each edited summary sentence, with the marked evidence as source context fed in the input.

**Fixing Factuality (FIX):** Given a factually incorrect summary sentence, edit it to make it factually correct, with reference to the source text. We create annotations using pre-edited summary sentence and the marked evidence as the input, and the post-edited sentence as the target.

**Topic-based Summarization (TOPIC):** Given the source article and a topic, the task is to generate a summary for a given topic from a source article. We use Wikipedia section headers as topics and select summary sentences from our labeled dataset that have evidence from a single section only. These sentences act as target summaries, while the full document and section header serve as input.

**Multi-sentence Compression (COMP):** Given a cluster of sentences from the source document, generate a single sentence summary that incorporates information from all of them. We create labeled data for this by using each summary sentence as a target and its marked evidence as the input.

**Evidence Extraction (EVEXT):** Given a source document and a summary sentence, identify a minimal set of source sentences which collectively provide supporting evidence for all facts present in that summary sentence. The labeled data consists of each summary sentence and the full source document as input, and the evidence links marked by annotators as the ground truth.

**Unsupported Span Prediction (UNSUP):** Given a summary sentence and a set of sentences from the source providing evidence, predict spans in the summary which are not supported by the evidence. To create labeled data, we select those summary

| Domain | Count | Domain | Count |
|--------|-------|--------|-------|
| Biographies | 1514 | Companies | 97 |
| Schools | 150 | Landmarks | 50 |
| Disasters | 145 | Newspapers | 32 |

Table 1: Number of annotated documents in various domain splits of our benchmark. Total 1988 documents across all domains.

sentences where annotators only made deletions (no additions or replacements). The input is the pre-edit summary sentence and the marked evidence, and the gold target is the set of spans that were deleted from the summary by annotators.

## 4 Dataset Overview and Statistics

The USB is a benchmark comprising 6 different domains with a varying number of instances in each (Table 1). We use a 40:20:40 split for train, validation and test set size for each domain, except the *landmarks* and *newspapers* domains due to small size. Articles from these two domains are kept as challenging test sets to measure the out-of-domain generalization. Length distributions of source documents and their summaries are shown in Figure 3 in the Appendix. Both exhibit long-tail distributions with lengthy sequences — about 32% of source documents have more than 2000 words and 10% of summaries have more than 200 words. We also find that 27% of summary sentences correspond to 4 or more marked evidence sentences (Figure 3 in the Appendix). This suggests a high degree of abstractiveness, because information needs to be combined from many source sentences and expressed in a single sentence. Annotators deleted about 22% of the words on average from the initial summary presented to them, while adding about 2% new words.

## 5 Benchmarking Different Models

We run a suite of models on all tasks in our benchmark and present the results in Table 2. For this set of experiments, we use the consolidated train, validation and test splits, which are a union of the corresponding splits from all domains. For tasks that involve generation of summaries, we use Rouge score (Lin, 2004) as the metric. We show geometric mean of the 1,2, and L variants for succinctness (Table 2). One exception is the Fixing Factuality task for which we use exact match as the metric. For Unsupported Span Prediction, we measure the F1 score based on BIO tagging format (Sang and

Buchholz, 2000). For the remaining tasks we use standard binary classification metrics.

For the classification/span prediction tasks in our benchmark, we fine-tune Roberta-Large (Liu et al. 2019; Table 2). We recast these as seq2seq tasks and fine-tune variants of T5 models on each of the 8 tasks. We include the original (Raffel et al., 2020) and the instruction-tuned Flan version (Chung et al., 2022). T5 Large outperforms Roberta-Large on all the classification/span prediction tasks. Flan-T5 Large performs similarly to T5 Large, though achieves notable gains on Unsupported Span Prediction. Flan-T5 XL consistently improves performance over larger models on almost all tasks, suggesting model size helps (Table 2). We also train a multi-task variant of Flan-T5-XL (on all tasks jointly). This mostly retains similar performance as a dedicated XL model trained only on that task, except for Evidence Extraction and Unsupported Span Prediction (Table 2).

We run large language models including publicly released models (for research purposes) such as Llama (Touvron et al., 2023) and Vicuna (Chiang et al., 2023), and closed models such as OpenAI's gpt-3.5-turbo[5], i.e., *ChatGPT*. For tasks where the full document is fed as input, we use 4 examples for few-shot prompting owing to limitations in the maximum feasible sequence length for these models, while for the rest we use 16 examples (for details, see the Appendix). ChatGPT consistently outperformed Vicuna-13B and Llama-13B on all tasks except Fixing Factuality. This is because for the Fixing Factuality task, ChatGPT almost always adds new unnecessary information to the summary, even after prompting it to not do that. Compared to ChatGPT, finetuned models perform better on every task. The performance difference is largest for factuality-based tasks such as Unsupported Span Prediction, Evidence Extraction, and Fixing Factuality. ChatGPT does comparatively well on tasks that involve generating summaries.

Since automatic metrics for measuring summary quality like ROUGE (Lin, 2004) do not necessarily mirror human preference (Cohan and Goharian, 2016), we conducted *human evaluation* of the generated summaries in the COMP, ABS and TOPIC tasks. We collect ratings for summaries generated by Flan-T5 and ChatGPT for 50 randomly selected documents from the test set, using a questionnaire (see the Appendix for more details). We found that

---

[5]We used the frozen version codenamed gpt-3.5-turbo-0301

| Model | COMP | EVEXT | EXT | FAC | FIX | ABS | TOPIC | UNSUP |
|---|---|---|---|---|---|---|---|---|
| Metric→ | Rouge | F1 | AUC | AUC | ExactMatch | Rouge | Rouge | F1 |
| Fine-tuned models | | | | | | | | |
| RoBERTa-Large | - | 71.01 | 84.06 | 92.69 | - | - | - | 49.21 |
| T5-Large | 41.97 | 77.22 | 87.00 | 94.89 | 31.26 | 33.44 | 23.81 | 51.71 |
| Flan-T5-Large | 43.23 | 77.71 | **87.99** | 95.15 | 32.94 | 32.05 | 23.62 | 58.57 |
| Flan-T5-XL | **44.87** | **79.23** | 87.81 | 95.30 | 35.10 | **32.69** | **24.26** | **64.94** |
| Flan-T5-XL (multitask) | 44.32 | 76.64 | 86.44 | **95.38** | **36.71** | 31.83 | 23.46 | 58.51 |
| Few-shot prompted LLMs | | | | | | | | |
| Llama-13B | 28.12 | 5.56 | 52.90 | 49.34 | 8.20 | 5.51 | 2.47 | 0.63 |
| Vicuna-13B | 31.35 | 6.65 | 52.76 | 55.28 | 4.28 | 5.56 | 2.84 | 1.47 |
| GPT-3.5-turbo | 33.21 | 26.78 | 61.63 | 60.81 | 3.29 | 29.77 | 14.59 | 7.80 |

Table 2: Performance of models on different tasks evaluated on the full test dataset. Tasks: **COMP**: Multi-sentence Compression **EVEXT**: Evidence Extraction **FAC**: Factuality Classification **FIX**: Fixing Factuality **ABS**: Abstractive Summarization (of full document) **EXT**: Extractive Summarization **TOPIC**: Topic-based Summarization **UNSUP**: Unsupported Span Prediction

on average, ChatGPT's summaries are mostly preferred over Flan-T5-XL model's summaries for all 3 summary generation tasks in terms of relevance and factuality (Table 3). This suggests that while fine-tuned models produce summaries closer to the ground truth in the dataset (thus achieving high ROUGE), humans may find the summaries of few-shot prompted LLMs better. For example, in the Topic-based summarization task, while Flan-T5-XL produces summaries with an average length of 46.3 words, ChatGPT generates summaries with an average length of 110.2 words. The ground truth summaries for that task are 36.9 words long on average, which is more closely matched by Flan-T5-XL, but the much longer summaries of Chat-GPT are considered better by human annotators as reflected in the human ratings (Table 3).

For the Fixing Factuality (FIX) task, we compare the *fixed* summaries generated by Flan-T5-XL and ChatGPT, asking which of them (i) remove more factual errors; (ii) mistakenly remove more correct information; (iii) add more new information; to the initially provided incorrect summary. We found that while ChatGPT removes more factual errors from summaries than Flan-T5, it often does so by removing lots of (even factual) information altogether, and replacing it with new content to effectively make a new different summary (Table 3).

## 6 Out-Of-Domain Performance on Tasks

We next evaluate the performance of fine-tuned models when tested on a domain different from what they were trained on. Our benchmark has training data from 4 domains (i.e. excluding *landmarks* and *newspapers*), with different amounts of labeled data for each. To control for training set size, we randomly subsample annotated documents for each domain to isolate 40, 19, and 38 documents for train, validation and test splits. These sizes of the splits were chosen to match the smallest of the 4 domains i.e. *companies* (Table 1).

We train and evaluate Flan-T5 Large models on different domains and plot average scores for all tasks training and test domain pair in Figure 2. Models trained on the same domain as the test domain perform best or negligibly close to it. But across test domains, the best out-of-domain trained model has $< 15\%$ performance drop compared to this, showing respectable average out-of-domain performance. Going by the in-domain performance of models trained on equal amounts of data, the *biographies* domain is the easiest and the *disasters* domain is the most difficult. One distinction between the *disasters* domain and others which might explain its difficulty is that it deals with summarizing an event rather than an entity.

For each task in our benchmark, we investigate whether having access to a large training dataset (irrespective of domain) is more important than having training data from the test domain. We use the test splits of 3 domains (*companies*, *disasters*, and *schools*), and on each of them we evaluate 3 different models trained on: (1) The training split of the same domain; (2) The training split of the

| Abstractive Summarization (ABS) | | |
|---|---|---|
| **Question** | **Flan-T5** | **GPT-3.5-turbo** |
| Which of the following summaries is better in terms of effectively summarizing the given full content? | 36.4% | 39.7% |
| Which of the following summaries is more factual, accurately representing the information presented in the given full content? | 33.8% | 33.1% |
| **Multi-sentence Compression(COMP)** | | |
| **Question** | **Flan-T5** | **GPT-3.5-turbo** |
| Which of the two summaries covers more information touching upon all the highlighted sentences? | 27.6% | 50.0% |
| Which of the following summaries is more factual, accurately representing the information presented in the document? | 21.1% | 38.8% |
| **Topic-based Summarization(TOPIC)** | | |
| **Question** | **Flan-T5** | **GPT-3.5-turbo** |
| Which of the two summaries is better in terms of effectively summarizing the given topic? | 10.0% | 85.3% |
| Which of the two summaries is more related to and exclusive to the given topic? | 11.3% | 77.3% |
| **Fixing Factuality(FIX)** | | |
| **Question** | **Flan-T5** | **GPT-3.5-turbo** |
| Which of the two summaries removes more contradictory/unsupported information from the incorrect summary, in reference to the context? | 18.0% | 38.0% |
| Which of the two summaries removes more correct information (which is actually well-supported by the context) from the incorrect summary? | 3.0% | 24.0% |
| Which of the two summaries adds more new facts compared to the incorrect summary? | 2.0% | 67.0% |

Table 3: Win rate for model outputs along different aspects as indicated in human evaluation for different tasks

| Training Domain | COMP | EVEXT | EXT | FAC | FIX | ABS | TOPIC | UNSUP |
|---|---|---|---|---|---|---|---|---|
| Metric→ | Rouge | F1 | AUC | AUC | ExactMatch | Rouge | Rouge | F1 |
| **Companies** | | | | | | | | |
| Companies | 30.02 | 61.61 | 66.36 | 90.10 | 11.76 | 19.30 | 18.51 | 7.41 |
| Biographies | -1.83 | +2.07 | +2.22 | -2.80 | -5.88 | -3.19 | -3.62 | -7.41 |
| Biographies (full) | +0.67 | +6.42 | +16.57 | +3.84 | +20.59 | +0.40 | -2.92 | +46.85 |
| **Disasters** | | | | | | | | |
| Disasters | 31.69 | 52.89 | 77.89 | 77.67 | 3.03 | 21.68 | 16.95 | 5.80 |
| Biographies | -2.75 | +7.15 | -9.84 | +6.91 | -1.01 | -5.31 | -1.54 | -5.80 |
| Biographies (full) | -2.09 | +12.52 | +6.36 | +12.55 | +15.15 | +0.45 | +0.78 | +40.22 |
| **Schools** | | | | | | | | |
| Schools | 38.63 | 62.72 | 73.92 | 88.89 | 3.19 | 28.89 | 25.04 | 2.70 |
| Biographies | -2.09 | -0.24 | -0.72 | -1.84 | +2.13 | -8.45 | -5.67 | +0.12 |
| Biographies (full) | +0.69 | +5.20 | +10.60 | +3.44 | +26.60 | -4.88 | -4.51 | +37.98 |

Table 4: Out-Of-Domain evaluation of fine-tuned Flan-T5-Large models. In each section of the table, we evaluate 3 variants - A) Model trained on the test domain (Companies, Disasters & Schools), B) Model trained on the Biographies domain (training sets of A and B are subsampled to have equal number of datapoints: train-40, validation-19, test-38), and C) Model variant trained on the full biographies dataset with 607 datapoints for training. Factuality related tasks benefit greatly from an abundance of training data, even if it's not from the target domain.

*biographies* domain, and; (3) The *full* training split of the *biographies* domain (before subsampling) which contains 607 annotated documents. Training on equivalent amounts of data from the test domain and biographies domain leads to comparable or worse performance of the latter (Table 4).

However, training on the full train set of the biographies domain achieves much higher perfor-

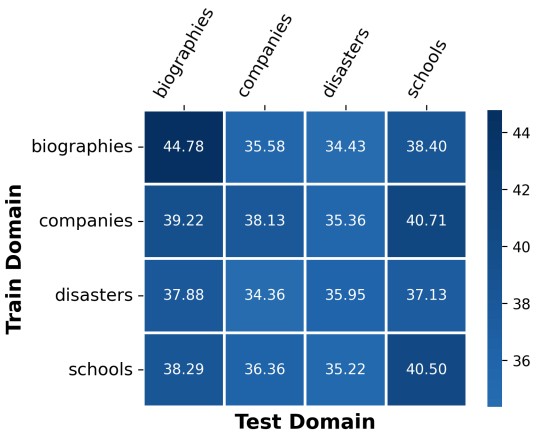

Figure 2: Average cross-domain model performance (using Flan-T5-Large) on benchmark tasks. All domains are subsampled to use equal number of annotated documents (train–40, validation–19, test–38).

mance on many tasks, despite the domain shift (Table 4). Gains are most visible on the Unsupported Span Prediction and Fixing Factuality tasks. By contrast, for tasks requiring summary generation, using the large biographies training set often does worse than using the $15\times$ smaller in-domain train set. This might happen because domain-specific knowledge is required to learn the style of summaries to generate for a given domain. On the other hand, factuality related tasks tend to be more objective (e.g., judging factual correctness), and so model skills are transferrable across domains.

## 7 Comparison with Heuristic Annotations

For some tasks in our benchmark, past works have used heuristics to create large labeled training data sets as an alternative to collecting manual annotations (Chen and Bansal, 2018; Kryściński et al., 2020; Balachandran et al., 2022). We use such proposed heuristics to train models and compare them with models trained on high-quality, human annotated data. We conduct experiments on the Evidence Extraction, Factuality Classification and Fixing Factuality tasks. Because the primary advantage of heuristic-generated training sets is their size, we also assess how smaller human-labeled training sets fare in comparison.

For the Evidence Extraction task, we use lexical overlap as a proxy to derive "reference" evidence alignments. For example, we select the source sentence with the highest ROUGE-L score with a summary sentence as its evidence, as outlined in Chen and Bansal (2018). We also create a training set variant where entity overlap is used instead of

| Evidence Extraction | |
| --- | --- |
| | **F1** |
| SuperPAL (Ernst et al., 2021) | 53.8 |
| ROUGE (Chen and Bansal, 2018) | 40.9 |
| Entity overlap | 47.0 |
| Human annotations 100% (N=765) | 77.7 |
| Human annotations 5% | 70.9 |
| **Factuality Classification** | |
| | **AUC** |
| FactEdit (Balachandran et al., 2022) | 74.6 |
| FactCC (Kryściński et al., 2020) | 68.9 |
| Human annotations 100% | 95.1 |
| Human annotations 5% | 90.4 |
| **Fix factuality** | |
| | **Exact Match** |
| FactEdit (Balachandran et al., 2022) | 1.0 |
| FactCC (Kryściński et al., 2020) | 0.8 |
| Human annotations 100% | 32.9 |
| Human annotations 5% | 11.2 |

Table 5: Comparing use of human annotations vs heuristic annotations for finetuning Flan-T5 Large models. We also report performance when finetuning on $5\%$ of the training set with human annotations.

ROUGE-L to derive references. Finally, we use *SuperPAL* (Ernst et al., 2021) as an out-of-the-box solution to predict evidence labels for our dataset's summaries, and then use them for model training.

To train models to detect or fix factual errors, we artificially introduce errors into summaries to be used as exemplars. We do this via transformations such as swapping entities, numbers, pronouns, introducing negation, and so on, inspired by prior work (Kryściński et al., 2020). To generate diverse errors and hallucinations, we follow Balachandran et al. (2022); we mask parts of the summary out and then use a language model to infill these with (mostly unsupported) information.

We train models for 3 tasks on both heuristically-generated and manually annotated training datasets, and evaluate them on clean human-labeled test sets. Training on human-annotated data performs better than all heuristic-based alternatives across all tasks (Table 5). Next, we train models on subsets of the manually annotated datasets with varying sizes and compare their performance on the test sets; this shows how even a little human-annotated data can outperform large amounts of heuristic-generated data for different tasks. For each of the three tasks, the performance achieved using only $5\%$ of the hu-

man annotated training set, still outperforms the heuristically labeled full training set (Table 5). This highlights the value in collecting manual annotations, even if in small quantities, over using heuristics to generate training data labels.

## 8 Related Work

The tasks in our benchmark have been studied in prior work to varying degrees. The greatest amount of attention has gone to the tasks of extractive summarization (Wong et al., 2008; Kågebäck et al., 2014; Zhang et al., 2016; Nallapati et al., 2017), and abstractive summarization (Liu and Lapata, 2019; Zhang et al., 2020; Raffel et al., 2020; Lewis et al., 2020; Goyal et al., 2022). There exist plenty of datasets for abstractive summarization (Narayan et al., 2018; See et al., 2017; Kim et al., 2019; Wang and Ling, 2016). However, many of them were created heuristically, with "targets" being automatically extracted via rules from documents pulled from the web. This can lead to poor quality reference summaries (Bommasani and Cardie, 2020; Krishna et al., 2022), and training on them can yield models prone to generating hallucinations (Nan et al., 2021; Ji et al., 2022). By contrast, we use manual annotation to ensure that summaries are fully supported by sources, resulting in a high quality abstractive summarization dataset.

Past works for predicting factual correctness of summaries incorporate question-answering models and natural language inference methods (Scialom et al., 2021; Fabbri et al., 2022; Goyal and Durrett, 2021), or use synthetically introduced factual errors (Kryściński et al., 2020) to train models. In contrast, the USB benchmark introduces a high-quality manually annotated dataset for predicting factual correctness. For the task of *editing* summaries to fix factual errors, datasets with both synthetic and model-generated errors have been created (Balachandran et al., 2022; Liu et al., 2022). The task of unsupported span prediction is akin to detecting hallucinated content in generated summaries, and to the best of our knowledge, no labeled dataset exists for this task.

For extracting evidence for a summary, past works have used lexical overlap based heuristics (Chen and Bansal, 2018; Lebanoff et al., 2019). A manually annotated dataset for the task was introduced by Ernst et al. (2021), albeit our work provides a substantially larger manually annotated dataset. Similarly, for multi-sentence compres-

sion we introduce a much larger manually labeled dataset than prior works (Slobodkin et al., 2022). Prior research has mostly approached topic based summarization by adopting a predefined set of topics (Krishna and Srinivasan, 2018; Akhtar et al., 2017; Hayashi et al., 2021). However, we did not restrict the set of topics in our dataset, resulting in a long tail of (potentially challenging) rare topics.

## 9 Conclusion

We introduced the USB benchmark comprising tasks to measure model performance across different text summarization sub-tasks. We showed that fine-tuned smaller models outperform few-shot prompting of much larger LLMs by a large margin on tasks related to appraising the factuality of summaries. We studied how fine-tuned summarization models perform on out-of-domain data, and identified several tasks where the training dataset size is more important than its domain.

Finally, we showed that rather than training models on large volumes of heuristically labeled data, one can get better performance by creating a much smaller ($\approx 20\times$ smaller) manually labeled training set instead. The resultant USB benchmark permits the training of models for useful tasks such as extracting evidence for a summary, correcting factual errors in it, and generating summaries focused on specific topics. Our hope is that this benchmark spurs further research on these tasks and will serve as a barometer for progress in them.

## 10 Acknowledgements

We gratefully acknowledge the National Science Foundation (RI 2211954, RI 2211955, and FAI 2040929), UPMC, Highmark Health, Abridge, Ford Research, Mozilla, the PwC Center, Amazon AI, JP Morgan Chase, the Block Center, the Center for Machine Learning and Health, and the CMU Software Engineering Institute (SEI) via Department of Defense contract FA8702-15-D-0002.

## 11 Limitations

Despite efforts to collect a diverse dataset, the benchmark used in this study may still exhibit certain biases. The sampling process and the selection of Wikipedia articles as the primary data source could introduce inherent biases, potentially affecting the generalizability of the results. These biases may stem from the specific domains or topics covered in the dataset, as well as the way in which

Wikipedia articles are written and formatted. The dataset's reliance on Wikipedia articles as the primary source of data might not adequately represent the nuances and challenges encountered in different domains or sources. One prominent example is conversations which are frequently used in summarization research but are not represented in the benchmark. Similarly, a model's ability to detect errors/hallucinations in summaries in the benchmark may not necessarily reflect its ability to detect errors more broadly in summaries generated by a variety of models.

While the benchmark dataset was annotated by human annotators, it is important to acknowledge the possibility of annotation errors or inconsistencies. Despite efforts to ensure high-quality annotations, the presence of errors should be taken into account when interpreting the results. Human annotation is subjective by nature, and different annotators may have varying interpretations in some situations, e.g., deciding whether a fact in the summary requires explicit evidence or should be presumed as common knowledge.

## 12 Ethics Statement

**Potential biases:** When selecting the pool of annotators on Amazon Mechanical Turk (AMT) for creating the dataset, we required their location to be the United States. This was done since the US has a very large population of native English speakers, which can help in getting high quality annotations. However, this geographical restriction can also lead to biases in the annotation process. For example, it would affect what's considered common knowledge when assessing evidence for summaries. An annotator from the United States would likely consider a person's birth in Los Angeles as evidence of them being from California, because they know Los Angeles is in that state. However, if it were some other city and state in a country unfamiliar to them, they may not make a similar inference.

**Compensation for annotators:** For the initial qualification task, workers were paid 2 USD. After selecting the qualified workers, for the main annotation task workers were paid 2 to 3 USD per document-summary pair, depending on the number of sentences in the summary and the domain where it came from (we observed that some domains were more difficult). For the second round for verification, we paid annotators between 0.3 to 1.0 USD depending on the number of sentences in

the summary which were flagged for verification, which can be as low as 1 sentence. The creation of the entire dataset costed about 6000 USD including platform fees paid to AMT and server hosting costs.

**Use of proprietary LLMs:** We included the GPT-3.5-turbo large-language-model from OpenAI in our experiments since it has demonstrated excellent performance on diverse NLP tasks in zero-shot and few-shot settings. Unfortunately, OpenAI could discontinue hosting the model in future at which point it may not be possible to reproduce its results on the tasks proposed in this work. For this reason we have also included results with public open-source LLMs like Llama and Vicuna, as these models are publicly available and hence their results can always be reproduced.

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

## A Appendix

### A.1 Sample datapoints for different tasks

We show a sample labeled datapoint for each task from the validation set of USB in Figure 5 and Figure 6.

### A.2 Instructions used in model inputs

We list the instructions used in the inputs to Flan-T5 models in Table 6, Llama-13B in Table 7, Vicuna-13B in Table 8, and GPT-3.5-turbo in Table 9.

### A.3 Implementation details for models

In this section we outline the architectures, and input/output formatting used for different models used in our experiments. Additionally, we report the hyperparameters used during training and inference for each model in Table 10.

**Roberta** For the Factuality Classification task, we feed in the evidence and summary separated by the SEP token into a standard classifier setup, which applies a linear layer with sigmoid activation on top of the CLS embedding. For Evidence Extraction, we use the same architecture and input individual pairs of a summary sentence with each source sentence to make a prediction for each of them. For the Extractive Summarization task, we use a hierarchical architecture identical to the one described as BERT-LSTM in Krishna et al. (2021), except that we use a Roberta encoder instead of BERT. For Unsupported Span Prediction, we frame it as a sequence tagging problem where the given summary sentence and evidence are passed through Roberta and a linear layer with sigmoid predicts whether each token is supported or not. The consecutive positive predictions are concatenated to turn them into spans.

**T5/Flan-T5** We preface each input with an instruction for the task to be done, followed by the text from the source/summary to be input. We frame the Evidence Extraction and Extractive Summarization tasks as a sequence of Yes/No predictions for each sentence in the source. Each source sentence in the input is prefixed by an enumerated sentence id (e.g. SENT34), and the ground truth target is the sequence of all sentence ids, with a Yes/No following each according to it's positive/negative label (e.g. "SENT0 Yes SENT1 No SENT2 No..."). Similarly, for Factuality Classifi-

cation, the target is a single Yes/No based on the label. During inference, we measure the probabilities of generated Yes/No tokens which allows us to measure AUC scores too. For Unsupported Span Prediction, we generated the ground truth target by surrounding the unsupported spans in the summary with begin-span and end-span tags.

**Llama/Vicuna** For Llama and Vicuna we use the exact same input formatting. Compared to the Flan-T5 data formatting, we use a different set of instructions for these models, after trying out plausible variants for each task on the validation set. We provide 4 different instances as few-shot examples following the instruction in each datapoint for each task. The few-shot examples are chosen by sampling from the training set without replacement. Due to limitations in sequence length, we only use a maximum of 2048 tokens for the few-shot examples. For the tasks which require the full document in the input (i.e. ABS, EXT, EVEXT, TOPIC), we use 4 examples with each having a maximum of 512 tokens. For the remaining tasks, we use 16 examples each with a maximum length of 128 tokens. The few-shot examples are sampled (without replacement) from the training set while creating the prompt for each datapoint in the test set. Since these are decoder-only models which essentially generate plausible completions of the input string, we preface each output with a word (e.g. "SUMMARY:", "LABELS:") in the few-shot examples and at the end of the prompt to trigger the generation of the required summary/labels.

**GPT-3.5-turbo** The formatting of input and output is exactly the same as for Llama/Vicuna for all tasks except Evidence Extraction and Extractive Summarization. For these two tasks, we found that this model performed much better if we prompted it to generate the source sentence ids which should be assigned the positive label, instead of generating a Yes/No prediction for each source sentence. So we changed the output formatting in our few-shot examples accordingly. For this model too, we choose a different set of instructions for the tasks by experimenting with different options on the validation set.

### A.4 Human evaluation of model outputs

It is well-acknowledged that ROUGE (Lin, 2004) is an imperfect automatic metric to assess summary quality, and may not accurately reflect human preferences (Nenkova, 2006; Cohan and Goharian,

2016; Goyal et al., 2022). Hence, we also conducted human evaluation for some tasks, where we show summaries generated by the best fine-tuned model (Flan-T5-XL) and the best fewshot-prompted LLM (GPT-3.5-turbo) and ask annotators to choose the better one along different dimensions (Table 3).

For the tasks of Abstractive Summarization (ABS), Multi-sentence Compression (COMP), and Topic-based Summarization (TOPIC), we collected annotations for 50 pairs of summaries, with 3 annotators rating each pair. For these 3 tasks, we did not screen workers based on qualification tasks since evaluating overall summary quality is a subjective task and it is better to have a diverse opinion from a large population, rather than a small set of manually selected people.

Evaluating model outputs for the Fixing Factuality (FIX) task is a more difficult but objective job. The increased difficulty comes from the need to carefully note the edits made by the models on the original incorrect summary and then decide on the factual validity and necessity of each edit. So we screened annotators via a qualification task on Mechanical Turk and selected 2 annotators to conduct the human evaluation for this specific task. Each pair of model outputs was rated by both annotators.

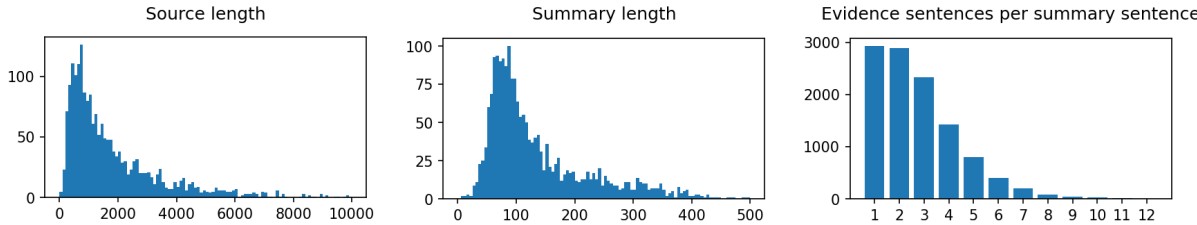

Figure 3: Distribution of number of words in the source and the summary, and the number of source sentences marked as evidence per summary sentence.

| Task | Instruction |
|---|---|
| Multi-sentence Compression (COMP) | Summarize the following content in a single line. |
| Abstractive Summarization (ABS) | Summarize the following content. |
| Fixing Factuality (FIX) | Rewrite the given summary of the content to make it factually correct. |
| Unsupported Span Prediction (UNSUP) | Annotate parts of the summary which are not supported by evidence from the content. |
| Topic-based Summarization (TOPIC) | Summarize the given content for the following topic. |
| Factuality Classification (FAC) | Is there sufficient evidence for the summary in the content? |
| Extractive Summarization (EXT) | For each sentence, predict if it is important. |
| Evidence Extraction (EVEXT) | For each sentence in the content, predict if it provides any evidence for the claim. |

Table 6: Instructions used in inputs to Flan-T5 models

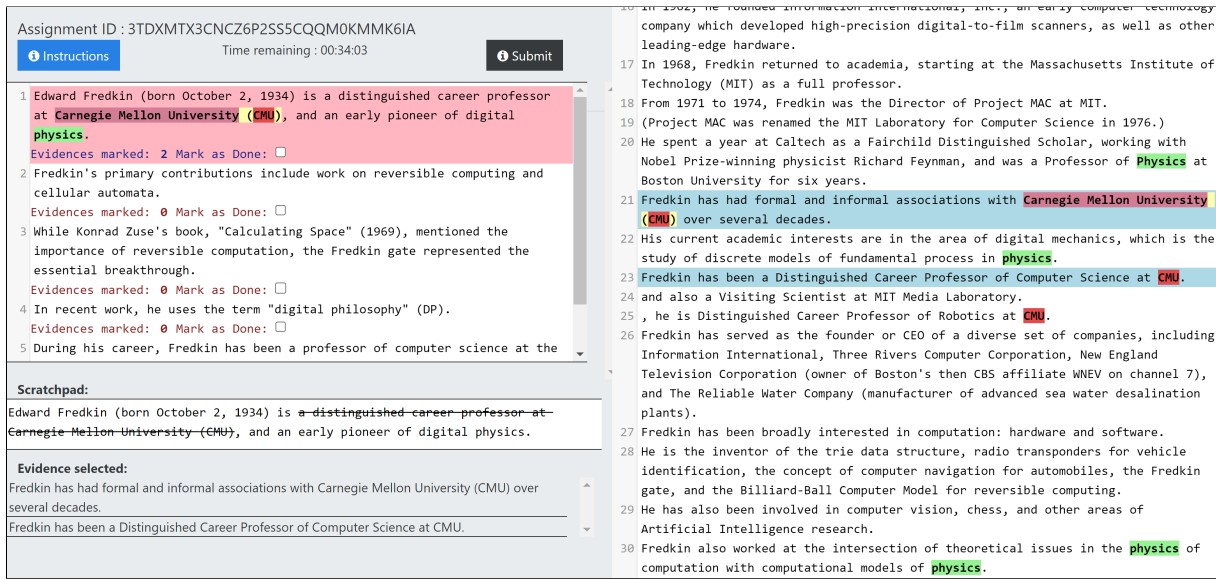

Figure 4: Screenshot of the interface used for collecting annotations. The summary is shown on the left and the source on the right. Entities in the active summary line are highlighted to help find evidence quickly. A scratchpad is provided where users can keep track of the parts of the summary for which evidence has been marked.

| Task | Instruction |
|------|-------------|
| Multi-sentence Compression (COMP) | Write a one-line summary of the content shown below. |
| Evidence Extraction (EVEXT) | Go over each sentence in the content, and decide if it supports the claim or not. Answer in Yes for a sentence if it supports the claim, and answer No otherwise. |
| Factuality Classification (FAC) | Is there sufficient evidence for the summary in the content? |
| Fixing Factuality (FIX) | Rewrite the given summary of the content to make it factually correct. |
| Abstractive Summarization (ABS) | Write a concise summary of the following paragraph |
| Topic-based Summarization (TOPIC) | Summarize the given content for the following topic. |
| Extractive Summarization (EXT) | For each sentence in the given content, label it as Yes if it is noteworthy enough to be included in a summary, or No otherwise. |
| Unsupported Span Prediction (UNSUP) | Regenerate the given summary, while surrounding those parts which do not have any supporting evidence in the content using [] and [/] tags |

Table 7: Instructions used in inputs to the Llama-13B model

| Task | Instruction |
|------|-------------|
| Multi-sentence Compression (COMP) | Write a single sentence summarizing the important points in the given content. |
| Evidence Extraction (EVEXT) | Predict which sentences in the given content can be used to infer facts in the claim. |
| Factuality Classification (FAC) | Decide if the following summary is consistent with the corresponding content. Note that consistency means all information in the summary is supported by the content. Explain your reasoning step by step then answer (yes or no) the question |
| Fixing Factuality (FIX) | Rewrite the following summary to make it factually accurate |
| Abstractive Summarization (ABS) | Draft a summary for the given document. |
| Topic-based Summarization (TOPIC) | Generate a summary of the given content covering the given topic. |
| Extractive Summarization (EXT) | For each sentence, predict if it is important. |
| Unsupported Span Prediction (UNSUP) | Annotate parts of the summary which are not supported by evidence from the content |

Table 8: Instructions used in inputs to the Vicuna-13B model

| Task | Instruction |
|---|---|
| Multi-sentence Compression (COMP) | Summarize the following content in a single line. |
| Evidence Extraction (EVEXT) | Below is a claim along with its corresponding content. Identify and list all the sentences within the content that partially or entirely support the claim. |
| Factuality Classification (FAC) | Decide if the following summary is consistent with the corresponding content. Note that consistency means all information in the summary is supported by the content. Answer yes or no. |
| Fixing Factuality (FIX) | The summary might be incorrect. How would you rewrite it to make it factually accurate? Make as little changes as possible. Do not add any new information to the summary. |
| Abstractive Summarization (ABS) | Draft a summary for the given document. |
| Topic-based Summarization (TOPIC) | Create a short summary of the given content that touches upon information which fall under the specified topic. |
| Extractive Summarization (EXT) | For the task of extractive summarization, list all the SENTs of the content which would be included in its summary. |
| Unsupported Span Prediction (UNSUP) | Go over the given summary carefully, and regenerate it while surrounding any parts which are not supported by the content using [] and [/] tags |

Table 9: Instructions used in inputs to the GPT-3.5-turbo model

| Model | Task | Learning rate | Batch Size | Max input length | Max output length |
|---|---|---|---|---|---|
| Roberta-Large | FAC | 1e-5 | 32 | 512 | - |
| Roberta-Large | EXT | 1e-5 | 32 | $128 \times 128^{\psi}$ | - |
| Roberta-Large | EVEXT | 2e-5 | 2048 | 128 | - |
| Roberta-Large | UNSUP | 2e-5 | 32 | 512 | - |
| T5-Large | (All) | 5e-5 | 32 | 8192 | 768 |
| FlanT5-Large | (All) | 5e-5 | 32 | 8192 | 768 |
| FlanT5-XL | (All) | 5e-5 | 64 | 1536 | 512 |
| Llama-13B | (All) | - | - | 6144 | 512 |
| Vicuna-13B | (All) | - | - | 6144 | 512 |
| GPT-3.5-turbo | (All) | - | - | Variable$^{\phi}$ | Variable$^{\phi}$ |

Table 10: Hyperparameters used for training and inference with different models. $\psi$: 128 sentences each with maximum of 128 tokens fed into a hierarchical model. $\phi$: GPT-3.5-turbo has a relatively small limit of 4096 tokens including both the input (with few-shot examples) and the output, and so we truncate the input on a per-task basis to leave token budget equal to the maximum output length in the train split for that task.

| Evidence Extraction | | | | | |
|---|---|---|---|---|---|
| | Accuracy | AUC | F1 | Precision | Recall |
| SuperPAL (Ernst et al., 2021) | 98.1 | 95.8 | 53.8 | 82.1 | 40.0 |
| ROUGE (Chen and Bansal, 2018) | 95.9 | 88.5 | 40.9 | 33.7 | 52.1 |
| Entity overlap | 95.7 | 92.5 | 47.0 | 35.6 | 69.4 |
| Human annotations 100% (N=765) | 98.8 | 99.0 | 77.7 | 77.0 | 78.4 |
| Human annotations 20% | 98.7 | 98.4 | 74.7 | 78.9 | 70.8 |
| Human annotations 10% | 98.5 | 98.1 | 72.4 | 73.0 | 71.8 |
| Human annotations 5% | 98.4 | 97.7 | 70.9 | 70.8 | 70.9 |
| **Factuality Classification** | | | | | |
| | Accuracy | AUC | F1 | Precision | Recall |
| FactEdit (Balachandran et al., 2022) | 55.7 | 74.6 | 29.4 | 72.6 | 18.4 |
| FactCC (Kryściński et al., 2020) | 52.9 | 68.9 | 20.1 | 66.3 | 11.8 |
| Human annotations 100% | 88.1 | 95.1 | 87.5 | 92.3 | 83.2 |
| Human annotations 20% | 86.7 | 93.9 | 86.1 | 90.6 | 82.0 |
| Human annotations 10% | 83.4 | 91.8 | 81.6 | 91.7 | 73.5 |
| Human annotations 5% | 82.6 | 90.4 | 82.5 | 83.2 | 81.7 |
| **Fix factuality** | | | | | |
| | Exact Match | Rouge-1 | Rouge-2 | Rouge-L | |
| FactEdit (Balachandran et al., 2022) | 1.0 | 81.6 | 73.0 | 81.0 | |
| FactCC (Kryściński et al., 2020) | 0.8 | 81.9 | 73.6 | 81.4 | |
| Human annotations 100% | 32.9 | 91.9 | 86.5 | 91.4 | |
| Human annotations 20% | 28.8 | 90.3 | 84.3 | 89.8 | |
| Human annotations 10% | 15.3 | 85.7 | 78.5 | 85.1 | |
| Human annotations 5% | 11.2 | 83.9 | 76.1 | 83.3 | |

Table 11: Comparision between using human annotations vs heuristic annotations for training models—Flan-T5-Large. We also report performance when finetuning on smaller fractions of the training set with human annotations.

| | | |
|---|---|---|
| Abstractive Summarization (ABS) | INPUT | **DOCUMENT:**
D'Vauntes Smith-Rivera
High school career
Smith-Rivera started high school at North Central High School in Indianapolis, and led his team to a state championship in his sophomore year.
He transferred to the basketball specialty Oak Hill Academy in Virginia for his senior year, and he helped lead the team to the 2012 national championship
He was recruited by Xavier, UCLA, Louisville, Memphis, NC State, and Georgetown.
… |
| | TARGET | D'Vauntes Smith-Rivera is a professional basketball player who last played for Koroivos of the Greek Basket League.
He played high school basketball for North Central in Indianapolis and Oak Hill Academy in Virginia.
… |
| Multi-sentence Compression (COMP) | INPUT | **SOURCE SENTENCES:**
Odenkirk was hired as a writer at "Saturday Night Live" in 1987 and worked there through 1991.
Odenkirk's friendship with Ben Stiller, with whom he briefly shared an office at "SNL", would lead to his being hired for the cast of "The Ben Stiller Show" in 1992.
Working as both a writer and actor on the show, he created and starred in the memorable sketch "Manson Lassie", and helped the show win an Emmy Award for writing. |
| | TARGET | From the late 1980s to 1990s, Odenkirk wrote for television shows "Saturday Night Live" and "The Ben Stiller Show", winning an Emmy Award for writing. |
| Extractive Summarization (EXT) | INPUT | **DOCUMENT:**
**SENT0:** D'Vauntes Smith-Rivera
**SENT1:** High school career
**SENT2:** Smith-Rivera started high school at North Central High School in Indianapolis, and led his team to a state championship in his sophomore year.
**SENT3:** He transferred to the basketball specialty Oak Hill Academy in Virginia for his senior year, and he helped lead the team to the 2012 national championship.
**SENT4:** He was recruited by Xavier, UCLA, Louisville, Memphis, NC State, and Georgetown.
… |
| | TARGET | SENT0 SENT2 SENT4… |
| Topic-based Summarization (TOPIC) | INPUT | **DOCUMENT:**
Arkema S.A.
Arkema was created when French oil major Total restructured its chemicals business.
The restructuring was a gradual process that began many years earlier:
…
**TOPIC NAME:** Organization |
| | TARGET | Arkema is organized into three business segments: Coating Solutions, Industrial Chemicals, and Performance Products. |

Figure 5: Sample input-output pairs for different tasks from the validation set of USB

| Factuality Classification (FAC) | INPUT | **EVIDENCE:**
In 2014 YG also expanded into the beauty industry with the creation of its cosmetics brand Moonshot.
YG Plus Inc., previously named Phoenix Holdings Inc., is a publicly traded media and advertising company acquired by YG Entertainment in November 2014.
**SUMMARY:**
In addition, the company operates a number of subsidiary ventures under a separate public traded company, YG Plus, which includes a clothing line, a golf management agency, and a cosmetics brand. |
| | TARGET | Incorrect |
| Unsupported Span Prediction (UNSUP) | INPUT | **EVIDENCE:**
David Martin McIntosh
McIntosh was born in Oakland, California, the son of Jean Marie (Slough), a judge, and Norman McIntosh.
He graduated with a B.A. (cum laude) in 1980, and later received a J.D. from University of Chicago Law School in 1983.
…
Incumbent Democrat U.S. Congressman Philip Sharp of Indiana's 2nd congressional district decided to retire.
McIntosh decided to run and won the Republican primary with a plurality of 43% in a four candidate field.
In the general election, he defeated Democratic Secretary of State of Indiana Joe Hogsett 54%-46%.
**SUMMARY:** David Martin McIntosh (born June 8, 1958) is an American attorney and Republican Party politician who served as the U.S. representative for Indiana's 2nd congressional district from 1995 to 2001. |
| | TARGET | David Martin McIntosh ~~( born June 8 , 1958 )~~ is an American attorney and Republican Party politician who served as the U.S. representative for Indiana 's 2nd congressional district ~~from 1995 to 2001~~ . |
| Fixing Factuality (FIX) | INPUT | **EVIDENCE:**
In 2009, Jordan returned to the F1 scene as a pundit for BBC Sport F1 coverage alongside Jake Humphrey (who was later replaced by Suzi Perry) and David Coulthard.
In March 2016 he was announced as Channel 4's lead analyst for C4F1.
**SUMMARY:**
He was the chief analyst for Formula One coverage on the BBC from 2009 to 2015 before joining Channel 4 after BBC pulled out in 2016. |
| | TARGET | He was the a pundit for Formula One coverage on the BBC from 2009 before joining Channel 4 in 2016. |
| Evidence Extraction (EVEXT) | INPUT | **DOCUMENT:**
**SENT0:** 2012 Istanbul rally to commemorate the Khojaly massacre
**SENT1:** "Justice for Khojaly" campaign.
**SENT2:** "Justice for Khojaly", or "JFK" for short, is an International Awareness Campaign, initiated on 8 May 2008 under the motto of "Justice for Khojaly, Freedom for Karabakh".
…
**SENT6:** Around 200,000 participants for the 20th anniversary remembrance of the Khojaly Massacre victims, dozens of youth and student organizations, public unions, Turkish organizations and movements participated in the rally.
…
**SENT17:** Various slogans included, "We are all from Khojaly", "Stop Armenian aggression", "Do not forget Turkic people genocide by Armenian gangs in southern Azerbaijan", "One nation, two countries, Justice for Khojaly!", and "Stop Armenian lies".
…

**SUMMARY:**
The demonstration with slogan "We are all from Khojaly" had around 200,000 participants. |
| | TARGET | SENT6 SENT17 |

Figure 6: Sample input-output pairs for different tasks from the validation set of USB