# OpenReview forum: "USB: A Unified Summarization Benchmark Across Tasks and Domains"
_EMNLP/2023/Conference — EMNLP 2023 Findings_

### Official Review · Reviewer_Ur5L · 2023-07-24

**Soundness:** 3

**Excitement:**

4: Strong: This paper deepens the understanding of some phenomenon or lowers the barriers to an existing research direction.

**Justification For Ethical Concerns:**

The only concern is the lack of an ethics consideration section to discuss the human annotation and evaluation detailedly, which correspondingly leads to the data quality concern.

In particular, the authors did not provide a detailed ethical consideration section that discusses the detailed human annotation and evaluation process. For example, what are the background of human annotators? Are they linguists or just native speakers? What are their education level? Are they compensated and how?

**Missing References:**

Some statements are not alway true.
L548-L554: "There exist plenty of datasets... many of them were created heuristically". However, *they are also many datasets that are human annotated*, for example, SAMSUM, QMSum, DialogSum. The authors could discuss more the difference compared with those abstractive datasets, instead of extractive ones.

Also, more recently, there is SummZoo benchmark, and more relevantly, there is MACSum, which contains multiple controllable attributes, and *is a unified benchmark as well*. However, they are not cited.


Zhang, Yusen, Yang Liu, Ziyi Yang, Yuwei Fang, Yulong Chen, Dragomir Radev, Chenguang Zhu, Michael Zeng, and Rui Zhang. "Macsum: Controllable summarization with mixed attributes." Transactions of the Association for Computational Linguistics 11 (2023): 787-803.

Zhong, Ming, Da Yin, Tao Yu, Ahmad Zaidi, Mutethia Mutuma, Rahul Jha, Ahmed Hassan et al. "QMSum: A New Benchmark for Query-based Multi-domain Meeting Summarization." In Proceedings of the 2021 Conference of the North American Chapter of the Association for Computational Linguistics: Human Language Technologies, pp. 5905-5921. 2021.

Gliwa, Bogdan, Iwona Mochol, Maciej Biesek, and Aleksander Wawer. "SAMSum Corpus: A Human-annotated Dialogue Dataset for Abstractive Summarization." In Proceedings of the 2nd Workshop on New Frontiers in Summarization, pp. 70-79. 2019.

Chen, Yulong, Yang Liu, Liang Chen, and Yue Zhang. "DialogSum: A Real-Life Scenario Dialogue Summarization Dataset." In Findings of the Association for Computational Linguistics: ACL-IJCNLP 2021, pp. 5062-5074. 2021.

Chen, Yulong, Yang Liu, Ruochen Xu, Ziyi Yang, Chenguang Zhu, Michael Zeng, and Yue Zhang. "Unisumm: Unified few-shot summarization with multi-task pre-training and prefix-tuning." arXiv preprint arXiv:2211.09783 (2022).

**Paper Topic And Main Contributions:**

This paper introduces a unified summarization benchmark (USB) that supports 9 summarization-related tasks.
To construct USB, the authors first conduct a sophisticated preprocessing pipeline to contain source text of their interest. Then, given a wikipedia article, the authors perform human annotation by asking human annotators 1) identify relevant evidence for each sentence in the summary (wikipedia leading paragraph); 2) edit (delete) contents in the summary, which cannot be supported by the wikipedia article (input text). In this way, they obtain parallel data, including article, leading paragraph, modified leading paragraph, supporting evidence, etc, and design 8 tasks using those data.
After building the USB, the authors conduct extensive experiments on it, and specifically analyze the OOD performance and influence of human annotated data compared with silver data.
Generally, this paper is well written and can provide some insights to the filed.

**Questions For The Authors:**

Q1 Instead of stating they are Mechanical Turk workers, could the authors provide more detailed ethical consideration that discusses the detailed human annotation process. For example, what are the background of human annotators? Are they linguists or just native speakers? What are their education level? Are they compensated and how?

Q2 How reliable are the human evaluation results? Have the authors measure their agreements (e.g. kappa)?

**Reasons To Accept:**

1. This paper is well-written and easy to follow.
2. The USB benchmark can serve as a valuable data source for future research on evidence mining and factuality.
3. Extensive experiments and detailed analysis.

**Reasons To Reject:**

Generally, this paper is well motivated and can be valuable to the filed.
The only concern is the lack of an ethics consideration section to discuss the human annotation and evaluation detailedly, which correspondingly leads to the data quality concern.

**Reproducibility:**

3: Could reproduce the results with some difficulty. The settings of parameters are underspecified or subjectively determined; the training/evaluation data are not widely available.

**Reviewer Confidence:**

4: Quite sure. I tried to check the important points carefully. It's unlikely, though conceivable, that I missed something that should affect my ratings.

---

> ### Author Rebuttal · Authors · 2023-08-29
>
> We appreciate your positive comments about the motivation and value of our paper.
> Regarding your concern about the absence of an ethics section, we recognize the importance of ethical considerations, especially when human annotators are involved. We will add an ethics section in the revised paper to discuss this in detail. For now we have mentioned the details below.
>
> ### Detailed Ethical Consideration for Human Annotation
> We exercised considerable care in the recruitment of workers for annotation. The workers were required to be from the United States to do our task. Workers needed to have a HIT approval rate of >95% and must have at least 1000 approved HITs. To ensure the expertise of the workers, we conducted an extensive qualification round where 174 annotators participated and 28 of them were selected by manually grading their responses by the authors. They were allowed to do the main task but even that their annotation quality was spot-checked, after about 15% of the dataset was collected, and 3 annotators were hand-picked because of their excellent annotation quality. The remaining 85% of the dataset was annotated by those 3 annotators. We also conducted a second round of verification where we had got part of the annotations re-checked by annotators again.
>
> Regarding worker compensation, for the initial qualification task, workers were paid 2 USD. After selecting the qualified workers, for the main annotation task workers were paid 2 to 3 USD per document-summary pair, depending on the number of sentences in the summary and the domain where it came from (we observed that some domains were more difficult to work with). For the second round for verification, we paid annotators between 0.3 to 1.0 USD depending on the number of sentences in the summary which were flagged for verification, which can be as low as 1 sentence.
>
>
> ### Agreement between raters for the human evaluation results
> Your question seems to pertain to the human evaluations comparing FlanT5 and ChatGPT (Table 5 in appendix). Below we show the agreement metrics among human evaluators on different tasks.
>
> For the multi-sentence compression (COMP), abstractive summarization (ABS) and topic-based summarization (TOPIC) tasks, each example was rated by 3 different raters, while for the Fix Factuality (FIX) task, two annotators rated each example.
> Below we show how often the raters agreed on the answer to the questions asked for each task:
>
> -----------------------------------------------------------------------------------------------------------------------
>
> Multi-sentence compression (COMP)
>
> Which of the two summaries covers more information touching upon all the highlighted sentences?
> All 3 raters agree: 46%
> Two raters agree: 52%
> Nobody agrees: 2%
>
> Which of the following summaries is more factual, accurately representing the information presented in the document?
> All 3 raters agree: 12%
> Two raters agree: 78%
> Nobody agrees: 10%
>
> -----------------------------------------------------------------------------------------------------------------------
>
> Abstractive summarization (ABS)
>
> Which of the following summaries is better in terms of effectively summarizing the given full content?
> All 3 raters agree: 38%
> Two raters agree: 52%
> Nobody agrees: 10%
>
> Consider the factuality of the summaries. Which of the following summaries is more factual, accurately representing the information presented in the given full content?
> All 3 raters agree: 20%
> Two raters agree: 70%
> Nobody agrees: 10%
>
> -----------------------------------------------------------------------------------------------------------------------
>
> Topic-based summarization (TOPIC)
>
> Which of the two summaries is better in terms of effectively summarizing the given topic?
> All 3 raters agree: 60%
> Two raters agree: 40%
> Nobody agrees: 0%
>
> Which of the two summaries is more related to and exclusive to the given topic?
> All 3 raters agree: 42%
> Two raters agree: 58%
> Nobody agrees: 0%
>
> ---------------------------------------------------------------------------------------------------------------------------
>
> Fixing Factuality (FIX)
>
> Which of the two summaries removes more contradictory/unsupported information from the incorrect summary, in reference to the highlighted text?
> Both raters agree: 76%
> Nobody agrees: 24%
>
> Which of the two summaries removes more correct information (which is actually well-supported by the highlighted context) from the incorrect summary?
> Both raters agree: 88%
> Nobody agrees: 12%
>
> Which of the two summaries adds more new facts compared to the incorrect summary?
> Both raters agree: 98%
> Nobody agrees: 2%

---

### Official Review · Reviewer_T3pT · 2023-08-04

**Soundness:** 3

**Excitement:**

3: Ambivalent: It has merits (e.g., it reports state-of-the-art results, the idea is nice), but there are key weaknesses (e.g., it describes incremental work), and it can significantly benefit from another round of revision. However, I won't object to accepting it if my co-reviewers champion it.

**Paper Topic And Main Contributions:**

The authors propose a unified summarization benchmark (USB). They use the overall section of Wikipedia pages as the candidate summary of the rest of the source. Then, they ask annotators to find the evidence for each sentence in the candidate summary and correct the sentences with no evidence. They propose 8 tasks over the annotated dataset and evaluate it on fine-tuned models and LLMs. Results show that fine-tuned models outperform few-shot LLMs. They also split the dataset into 6 domains and evaluated the cross-domain performances.

**Questions For The Authors:**

- QA line 80, if evidence was lacking, do you directly remove all sentences? Or do you remove only some related spans?
- QB line 205, do you only consider hyperlinks as entities? Have you ever done an experiment to show its coverage of all entities?
- QC line 248, how do you deal with the sentences that are verified to be wrong?

**Reasons To Accept:**

1. It is important to create a manually labeled summarization benchmark that includes various tasks. Especially for detecting the factual issues of the LLMs.
2. Overall, the paper is well-written and easy to follow. I can understand the motivation for each step of dataset construction and experiment.
3. The proposed benchmark is relatively comprehensive. It covers 8 tasks and 6 domains with an evaluation of two types of SOTA models. A number of insightful findings are proposed by analyzing experimental results.

**Reasons To Reject:**

1. Quality of the summaries is unknown. Although there is verification for annotation of the evidence, it is not clear about the quality of the summaries themselves. How could we know that the overview section of Wikipedia can serve as a good summary of the rest of the source text? Probably, some key information in the content sections is missing as a summary. The overall section is not designed to serve as a summary initially. That's why we can find factual errors in it. So I think the authors also need to prove they can serve as summaries by some human annotation.

2. Some proposed tasks are not natural. Since the dataset is not written but edited by humans, some tasks seem pseudo data as well. For instance, for EXT task, evidence is just for the overall section which may not be a good extractive summary. And for the TOPIC task, the topic-based summary is directly extracted from the overall section. I believe the readability will decrease. It is also a pseudo-summary rather than a human-written one. Similarly, in the COMP task, some coreference issues may happen. And it may not be a good summary of evidence without rewriting.

3. Domain bias. Biographies contain 1514 samples which is much larger than the sum of the rest domains. The second largest domain only contains 150 samples. This may weaken your conclusions on the domain experiments.

Overall, although some manual labels are involved, the construction of the dataset still contains some bias and noises which weaken the conclusions. Among the 8 tasks, the most useful or clean ones are fact-check-related tasks.




**Reproducibility:**

5: Could easily reproduce the results.

**Reviewer Confidence:**

4: Quite sure. I tried to check the important points carefully. It's unlikely, though conceivable, that I missed something that should affect my ratings.

**Typos Grammar Style And Presentation Improvements:**

For figure 2, I suggest to use 30 as pure white and 50 as pure blue to make the figure more clear.

---

> ### Author Rebuttal · Authors · 2023-08-29
>
> We appreciate your recognition of the importance of our work and its contributions to the field of summarization, particularly in detecting the factual issues of LLMs.
>
> ### Considering the introduction section of article as summary
> The official Wikipedia manual of (writing) style (https://en.wikipedia.org/wiki/Wikipedia:Manual_of_Style) dictates that “An article's content should begin with an introductory lead section – a concise summary of the article”;  and hence we believe this is a good summary of the topic of the article covering the most important information, whereas the long remaining part of the article goes into more details. The only issue is that the introduction section may also contain things not present in the remainder of the article, in which case it might be considered “unfaithful” to the full article. due to which it could not be considered a valid summary of the remaining article. To address exactly this, we employed annotators to edit the introduction section and remove information that’s not supported by the remainder of the article, thus creating properly aligned document-summary pairs.
>
>
> ### Ground truth for extractive summarization task (EXT) is not natural
> Given the abstractive summary of an article, we believe it is logical that the parts of the article that provide evidence for the summary should collectively be a good extractive summary of the article. This is because all the information which is present in the abstractive summary is also present in the corresponding evidence sentences from the source.
>
> ### Domain Bias
> The skew in domain samples is due to the difference in the amount of available data in Wikipedia (there are many more articles in the biographies category than the others). We specifically included rare domains to perform low-resource domain transfer experiments, as highlighted in Table 3 of our paper.
>
> ### Answers to more specific questions
>
> (Line 80) Only the unsupported span is removed. In some cases, the entire sentence may be removed if it is entirely unsupported.
>
> (Line 205) We considered only hyperlinks as entities to ensure a high level of certainty, as this information would be sent to human annotators. This approach was chosen to be time- and cost-efficient.
>
> (Line 248) Human annotators are instructed to remove parts of the summary sentence for which no evidence is present, and otherwise annotate unmarked evidence where it exists.

---

### Official Review · Reviewer_mh1S · 2023-08-09

**Soundness:** 3

**Excitement:**

3: Ambivalent: It has merits (e.g., it reports state-of-the-art results, the idea is nice), but there are key weaknesses (e.g., it describes incremental work), and it can significantly benefit from another round of revision. However, I won't object to accepting it if my co-reviewers champion it.

**Paper Topic And Main Contributions:**

In this paper, a benchmark derived from human-annotated Wikipedia content is presented, encompassing eight closely connected tasks: extracted summarization, abstractive summarization, topic-based summarization, sentence compressing, evidence selection, factual accuracy predicting, unsupported span prediction, factual errors correction. The authors also conduct a comprehensive comparison between traditional fine-tuned models and the most recent LLMs across these tasks, using both automatic and human evaluations.

**Questions For The Authors:**

The Landmarks and Newspapers are not included in any experiments?

**Reasons To Accept:**

The paper is well written. High quality summarization datasets are always important for the research community. This paper integrated eight very interesting summarization-related tasks and annotated a relatively large amount of examples for training and evaluation. This will be helpful for everyone in the research community.

**Reasons To Reject:**

My main worry is the absence of human assessment for the ultimate version of the proposed dataset. Maynez et al. 2020 noted that ground truth summaries are prone to hallucinations. I believe that incorporating thorough human evaluation of the dataset would provide researchers with confidence to explore this dataset extensively.
Due to the lack of human confirmation, I currently have some doubts about the quality of the annotated summaries. This is because, in Table 2, the difference in performance between ChatGPT and the fine-tuned models on generation tasks, as assessed w.r.t. the reference summaries, is quite substantial.
This indicates that summaries generated by fine-tuned models are more aligned with the annotated ground truth summaries, whereas summaries produced by ChatGPT do not closely resemble the ground truth. Interestingly, human evaluation highlighted a preference for summaries generated by ChatGPT. This suggests that the proximity to human-annotated summaries may not necessarily correlate with higher human satisfaction. Perhaps it's necessary to conduct a more thorough analysis of the distinctions between human-annotated summaries and the summaries generated by ChatGPT.

Additionally, the paper lacks an in-depth examination of the experimental outcomes. Of particular note is the notably poor performance of LLMs on tasks other than generation (such as EVEXT, EXT, FAC, UNSUP).

**Reproducibility:**

3: Could reproduce the results with some difficulty. The settings of parameters are underspecified or subjectively determined; the training/evaluation data are not widely available.

**Reviewer Confidence:**

4: Quite sure. I tried to check the important points carefully. It's unlikely, though conceivable, that I missed something that should affect my ratings.

---

> ### Author Rebuttal · Authors · 2023-08-29
>
> Thank you for recognizing the value of our work in contributing a high-quality, multifaceted benchmark for summarization tasks. We appreciate your positive comments on the paper's writing, the importance of quality datasets, and the range of tasks our benchmark supports.
>
> ### Absence of Human Assessment for the Ultimate Version of the Proposed Dataset
> We understand your concerns about the absence of human evaluations for the final version of the dataset. We can break down the quality of the dataset into 2 aspects: (1) the summary must have good coverage of the important aspects of the source, and (2) everything in the summary should have corresponding evidence in the source. While it is impossible to achieve perfection in both these criteria for any dataset, we have designed our dataset collection process to ensure them as follows :
>
> (1) We took the introduction section of the  Wikipedia article as the summary which covers most of the important content in the remaining article according to the official Wikipedia manual of (writing) style (https://en.wikipedia.org/wiki/Wikipedia:Manual_of_Style) : “An article's content should begin with an introductory lead section – a concise summary of the article”;  and hence we believe this is a good summary of the topic of the article covering the most important information, whereas the long remaining part of the article goes into more details.
>
> (2) We explicitly tasked annotators with editing summaries to remove unsupported facts from the source, which is rarely done in most widely used summarization datasets such as CNN-DailyMail, XSum, Reddit-TIFU etc. To ensure good quality of annotations we eventually selected the best 3 annotators out of a total 174 initial candidates based on manual assessment of their work. Even after all documents were annotated, we again ran a new task on Mechanical Turk to verify the annotations.
>
> Since the notion of a good summary varies from one person to another, a human assessment of the appropriateness of summary content is unlikely to reveal any meaningful results. This is further complicated by the fact that there is no reference point to compare the scores given by humans in such an evaluation.
>
> Similarly there is subjectiveness associated with deciding if any given part in the summary is unsupported or not. This is because often
> some “background knowledge” is required to infer the summary from the source (we talk about this in lines  642-652).  For example, if the source mentions that a person was born in Los Angeles, and the summary says that they are American, then annotators who know Los Angeles is in that state would consider that as sufficient evidence. However, another annotator who does not have that background knowledge might consider it an unsupported fact and rate it as a bad summary.
>
> ### Good human ratings of ChatGPT despite low ROUGE score
> Prior work has made very similar observation that ChatGPT summaries are preferred to that of fine-tuned models’ summaries by humans on popular benchmarks such as XSum, CNN-Dailymail (Goyal et al 2022, "News summarization and evaluation in the era of GPT-3”). This does not necessarily suggest that the ground truth summaries are low quality, but rather suggests that their form (e.g. length, choice of important content) is different than that of ChatGPT summaries (leading to low ROUGE scores). Humans prefer ChatGPT summaries, which is in line with observations made in prior literature.
>
>
> ### Poor performance of LLMs on factuality-related tasks
> We experimented with 3 different LLMs and tried extensive prompt-tuning to try and improve performance of fewshot prompting but the performance stayed far below that of finetuned models. This simply shows that fewshot prompting is not enough to teach models to do factuality related tasks. There is no reason why a model should be capable of learning any arbitrary task from a small number of demonstrations, and perhaps these tasks are ones where it is necessary to train the model with thousands of examples to teach them.
>
>
> ### Landmarks and Newspapers in Experiments
> The Landmarks and Newspapers datasets were not included in the training phase due to their small size. Instead, they were used only as challenging Out-Of-Domain (OOD) examples in the testing phase to evaluate the model's generalizability.

---

### Official Review · Reviewer_WFLP · 2023-08-11

**Soundness:** 3

**Excitement:**

3: Ambivalent: It has merits (e.g., it reports state-of-the-art results, the idea is nice), but there are key weaknesses (e.g., it describes incremental work), and it can significantly benefit from another round of revision. However, I won't object to accepting it if my co-reviewers champion it.

**Paper Topic And Main Contributions:**

They present a benchmark based on Wikipedia, enhanced with a comprehensive set of crowd-sourced annotations, supporting 8 interconnected tasks: (i) extractive summarization, (ii) abstractive summarization, (iii) topic-centered summarization, (iv) compressing selected sentences to a single line summary, (v) identifying evidence backing a summary, (vi) determining a summary's factual accuracy, (vii) spotting unsupported segments in a summary, and (viii) rectifying factual errors in summaries. When comparing different approaches on this benchmark, they find that medium-sized fine-tuned models consistently surpass larger few-shot prompted models across several tasks. Regarding tasks tied to factuality, heuristics-based training data perform worse than training on much fewer human-labeled datasets. Their sourced articles span 6 fields, enabling cross-domain studies. Depending on the task, the quantity of training data can be more influential than its domain origin, but for some tasks, domain-specific training, even if scarce, proves more effective.

**Questions For The Authors:**

could you please give more details about bout the annotators and how human build the dataset? More evidences for the quality of the datasets could be provided.

**Reasons To Accept:**

1. The benchmark offers a diverse platform for training and assessment on 8 unique tasks focusing on essential yet overlooked facets of text summarization.
2. Various models and training approaches, such as fine-tuning, few-shot prompting, and multi-task training, were evaluated.
3. They provided insights on how different tasks generalize outside their original domain, pinpointing which tasks prioritize the volume of training data over its specific domain origin.
4. The paper is well-written.

**Reasons To Reject:**

1.The dataset is not written but edited by humans, and the quality of the dataset is unknown. Therefore, the evaluations done on this dataset might not be so trustworthy.
2. The experimental analysis lacks depth, so the findings based on using this benchmark is not exciting enough.



**Reproducibility:**

4: Could mostly reproduce the results, but there may be some variation because of sample variance or minor variations in their interpretation of the protocol or method.

**Reviewer Confidence:**

3: Pretty sure, but there's a chance I missed something. Although I have a good feel for this area in general, I did not carefully check the paper's details, e.g., the math, experimental design, or novelty.

---

> ### Author Rebuttal · Authors · 2023-08-29
>
> We appreciate your positive comments on the uniqueness of the tasks, the evaluation of various models, and the insights provided by our paper.
>
> ### Quality of the Dataset
> To address the point that the summaries are not written but edited by human, we note that the introduction part of the Wikipedia articles even before the edits is actually a summary written collaboratively by humans (the many contributors who authored the Wikipedia page). The official Wikipedia manual of (writing) style (https://en.wikipedia.org/wiki/Wikipedia:Manual_of_Style) dictates that “An article's content should begin with an introductory lead section – a concise summary of the article”; hence we believe this is a good summary of the topic of the article covering the most important information, whereas the long remaining part of the article goes into more details. The only issue is that the introduction section may also contain things not present in the remainder of the article, in which case it might be considered “unfaithful” to the full article. To address this, we employed annotators to edit the introduction section and remove information not supported by the remainder of the article, thus creating properly aligned document-summary pairs.
>
>
> We employed rigorous quality assurance measures outlined as follows:
> We conducted rigorous screening of annotators via an initial qualification task which was attempted by 174 AMT workers and only 16% of them passed after manual grading by us.
> During the main task, we further monitored the annotations submitted by the workers and removed the ones whose annotations weren’t up to the mark, retaining only the best annotators.
> Finally, after all documents were annotated, we again ran a new task to verify the annotations. We used a trained model to identify summary sentences with potentially missing evidence, and got them re-checked by annotators who made the appropriate edits if there was a legitimate error in the prior annotation.
>
>
> ### Lack of Depth in Experimental Analysis
> We feel we have performed a robust set of experiments (though would be happy to add additional, specific experiments if suggested). To reiterate, we performed 3 sets of experiments.
>
> (1) We benchmarked several latest models for the tasks introduced and compare their performance especially comparing fine-tuning and few-shot prompting based approaches.
>
> (2) We evaluated the performance of fine-tuned models when tested on a domain different from what they were trained on.
>
> (3) We demonstrated the usefulness of our annotations by comparing against heuristic annotations done using previously proposed approaches.
>
> ### More details about the annotators
> In the Mechanical Turk interface, we filtered workers by the following criteria: (1) they should be located in the United States, (2) they need to have a HIT approval rate of >95%, and (2) they must have at least 1000 approved HITs. To ensure the expertise of the workers, we conducted an extensive qualification round where 174 annotators participated and 28 of them were selected by manually grading their responses by the authors. They were allowed to do the main task but their annotation quality was spot-checked, after about 15% of the dataset was collected, and 3 annotators were hand-picked because of their excellent annotation quality. The remaining 85% of the dataset was annotated by those 3 annotators.

---

### Meta-Review · Area_Chair_BPEe · 2023-09-17

**Recommendation:** 3

**Metareview:**

The paper introduces a Unified Summarization Benchmark (USB) encompassing 8 interconnected summarization tasks based on Wikipedia content, with a focus on factuality and correctness. The authors compare fine-tuned models and large language models (LLMs) on these tasks, revealing that fine-tuned models outperform LLMs, particularly in tasks tied to factuality. However, concerns regarding the quality of the human-annotated dataset and the need for more extensive human evaluation are raised, emphasizing the importance of assessing the dataset's reliability. Additionally, the paper lacks an in-depth analysis of experimental outcomes, especially the notable performance gaps observed in tasks other than generation. Some issues were resolved during the response period, specifically on human annotation.

---

### Decision · Program_Chairs · 2023-10-07

**Decision:**

Accept-Findings

**Comment:**

The paper introduces a Unified Summarization Benchmark (USB) encompassing 8 interconnected summarization tasks based on Wikipedia content, with a focus on factuality and correctness. The authors compare fine-tuned models and large language models (LLMs) on these tasks, revealing that fine-tuned models outperform LLMs, particularly in tasks tied to factuality. However, concerns regarding the quality of the human-annotated dataset and the need for more extensive human evaluation are raised, emphasizing the importance of assessing the dataset's reliability. Additionally, the paper lacks an in-depth analysis of experimental outcomes, especially the notable performance gaps observed in tasks other than generation. Some issues were resolved during the response period, specifically on human annotation.